# Reciprocating RNA Polymerase batters through roadblocks

Jin Qian [1], Allison Cartee [1], Wenxuan Xu [1], Yan Yan [1], Bing Wang [2], Irina Artsimovitch [2], David Dunlap [1] & Laura Finzi [1] ✉

RNA polymerases must transit through protein roadblocks to produce full-length transcripts. Here we report real-time measurements of *Escherichia coli* RNA polymerase passing through different barriers. As intuitively expected, assisting forces facilitated, and opposing forces hindered, RNA polymerase passage through *lac* repressor protein bound to natural binding sites. Force-dependent differences were significant at magnitudes as low as 0.2 pN and were abolished in the presence of the transcript cleavage factor GreA, which rescues backtracked RNA polymerase. In stark contrast, opposing forces promoted passage when the rate of RNA polymerase backtracking was comparable to, or faster than the rate of dissociation of the roadblock, particularly in the presence of GreA. Our experiments and simulations indicate that RNA polymerase may transit after roadblocks dissociate, or undergo cycles of backtracking, recovery, and ramming into roadblocks to pass through. We propose that such reciprocating motion also enables RNA polymerase to break protein-DNA contacts that hold RNA polymerase back during promoter escape and RNA chain elongation. This may facilitate productive transcription in vivo.

A dense array of proteins relevant to structure and function are associated with genomic DNA. These DNA-binding proteins play regulatory roles in cellular processes, such as genome packaging, and the recruitment and modulation of processive enzymes for replication and transcription[1–3]. These regulatory DNA-binding proteins vary significantly in their affinity for specific or non-specific DNA sequences, which may further change with physiological conditions[4]. Therefore, a motor enzyme, such as RNA polymerase (RNAP), encounters both low- and high-affinity DNA-bound roadblocking proteins. For uninterrupted transcription, these roadblocks must be transiently displaced during passage, whereas stalled RNAPs may require termination factors or convoys/collisions between polymerases to clear a template[5,6].

Although roadblocks that interfere with transcription have been studied for decades, how RNAP surpasses a roadblock is known only for specific cases[7]. In principle, RNAP might passively wait for the roadblock to spontaneously dissociate, or it might actively dislodge the roadblock. Previous studies have shown that, when an elongation complex (EC)

encounters an obstacle, such as a DNA lesion or a DNA-bound protein, it slides backwards[8]. In a backtracked EC, the nascent RNA occludes the active site, blocking nucleotide addition. The arrested EC can be rescued by a trailing RNAP, a translating ribosome, a DNA translocase such as Mfd, which pushes RNAP forward, or by Gre factors, which facilitate the RNAP-mediated cleavage of the nascent RNA to restore the RNA 3′ OH in the active site[9–11]. The stability of the backtracked EC at a roadblock can also modulate the probability and/or rate of passage[12]. These findings were mostly based on run-off transcription assays that do not reveal the dynamics of roadblocked RNAP.

To reveal these dynamics, and expose general principles underlying RNAP progress through roadblocks, we used magnetic tweezers to monitor *E. coli* EC progress on DNA templates with either of two roadblock proteins, *lac* repressor (LacI) bound at sites (also called operators) with different affinities for the protein, or the mutant endonuclease EcoRI Q111, which binds but does not cut DNA. The experiments were conducted with up to 5 pN forces opposing or assisting RNAP translocation with or without GreA, the major

[1]Physics Department, Emory University, Atlanta, GA, USA. [2]The Center for RNA Biology and Department of Microbiology, The Ohio State University, Columbus, OH, USA. ✉e-mail: lfinzi@emory.edu

backtracking resolution factor in *E. coli*. Our results indicate that RNAP can employ both passive and active mechanisms to overcome obstacles and that force and GreA modulate the effective pathway. We propose a model that well explains these observations and reveals alternative mechanisms for ECs transiting through roadblocks.

## Results

### GreA and forces opposing or assisting RNAP translocation change pausing at roadblocks

Pauses at roadblocks during transcription elongation were measured using digoxigenin-end-labeled DNA templates containing a T7A1 promoter, a binding site for either the LacI protein: Os ($K_d = 10$ pM), O1 ($K_d = 0.05$ nM) or O2 ($K_d = 0.1$ nM), or the EcoR1 Q111 protein ($K_d = 5$ pM), and a $\lambda$T1 terminator (Fig. 1A)[13–15]. Biotin-labeled RNAP holoenzyme coupled to a streptavidin-coated magnetic bead was introduced in flow chambers containing tethered DNA templates and manipulated in a magnetic tweezer microscope (Fig. 1B). Whether force opposed or assisted transcription depended on which end of the template was digoxigenin-labeled and fixed to the glass, while the magnitude of the external force applied to RNAP was set by the separation between the permanent magnets above the flow cell.

Halted ECs were prepared as described in the Methods section. RNA synthesis was restarted by the addition of all four NTPs (1 mM) at calibrated forces of 0.2, 0.7, 2.0, or 5.0 pN (Methods, Supplementary Fig. 1A). These forces would generate from sub-kT of ~ 2kT of energy on a ratchet-like transcribing RNAP, well within the physiological range of energy buffeting chromosomal DNA and the associated proteins[16]. In the presence of LacI, ECs paused near the operator site, where LacI was expected to bind, but eventually transited through the roadblock (Fig. 1C)[17]. Different beads, due to variations in size and iron oxide content, exert slightly different forces producing different tether extensions. Thus, individual records of transcription (tether length *versus* time) were shifted and scaled to align them, and a step-wise fitting algorithm was applied to produce a dwell-time histogram with distinct peaks at significant pauses and roadblocks (Supplementary Fig. 2). The roadblock-associated pauses were significantly longer than ubiquitous, random pauses and were identified as occurring within ± 20 nm (60 bp) of the roadblock binding site. A minor population (<10%) of traces had no roadblock-associated pauses, likely due to incomplete binding of roadblock proteins to the DNA tethers. Pauses shorter than 20 seconds were treated as ubiquitous pauses and were excluded from the analysis. To confirm that roadblock protein re-binding does not affect the observed pause times, control experiments were performed in presence of heparin, which would be expected to sequester unbound and dissociated roadblock proteins in solution (Supplementary Fig. 3).

In the presence of LacI, we used DNA templates containing one of three *lac* repressor binding sites Os, O1 and O2, listed here in order of decreasing affinity (see above). For templates containing the O1 or O2 sites, nearly all ECs successfully transited through the roadblock within one hour. The pause times were distributed exponentially under all conditions (Fig. 2A, C), except for LacI-O2 with assisting force, which will be discussed below, and the lifetimes changed with the direction (assisting *versus* opposing), but not the magnitude, of force (Fig. 2B, D). As expected, the intermediate affinity LacI-O1 roadblocks produced longer pause times than the lowest affinity LacI-O2 roadblocks. On templates containing an artificial, symmetric binding site Os that binds LacI with the highest affinity, a fraction of ECs paused indefinitely at roadblocks (Fig. 3A). Therefore, we determined both pause lifetimes (Fig. 3B) and the fraction of RNAPs passing through the LacI-Os roadblock (Fig. 3C). As for the pauses measured for LacI-O1/O2 roadblocks, pause times and passage ratios at LacI-Os roadblocks changed with the direction of force, whereas the magnitude of the force had a negligible effect.

Backtracking by ECs has been widely associated with roadblocked transcription[9,10]. Force that assists EC translocation may prevent backtracking or favor recovery from backtracked states. Similarly, opposing force may promote backtracking or prevent recovery from backtracked states. To reveal the contribution of backtracking to pausing at roadblocks, GreA was added to rescue backtracked ECs by promoting the cleavage of the 3′ end of nascent RNA obstructing the active site[18]. GreA did not change pauses under assisting force (Figs. 2B, D and 3B), most likely because EC backtracking, and therefore GreA activity, were negligible under assisting-force conditions. On the contrary, GreA apparently increased the rate of recovery from backtracked states to accelerate passage through roadblocks under opposing force (Figs. 2B, D and 3B). This is consistent with previous work showing that GreA enhanced passage through LacI roadblocks[19]. Remarkably, unlike sequence-induced backtracking[20], even a very gentle force of 0.2 pN significantly reduced backtracking by roadblocked ECs. This is consistent with the low energy barriers to backtracking found previously for Rpo41 and PolII[21].

### GreA and tension reveal two paths through roadblocks

The direction of force and the addition of GreA changed pausing at LacI bound to O1, O2, or Os operators differently. For DNA templates containing either O1 or O2, opposing force lengthened pauses relative to the assisting-force baseline, but adding GreA restored that baseline (Fig. 2B, D). These results support an accepted notion that backtracking, without recovery promoted by Gre factors, inhibits productive RNA synthesis. However, on templates containing Os, opposing force without GreA shortened the pauses to a level below that observed under assisting force, and the effect was enhanced by the addition of GreA (Fig. 3B). The unexpected stimulatory effect of opposing force suggested that relatively fast cycles of backtracking and subsequent recovery, a reciprocating motion, may enable RNAP to

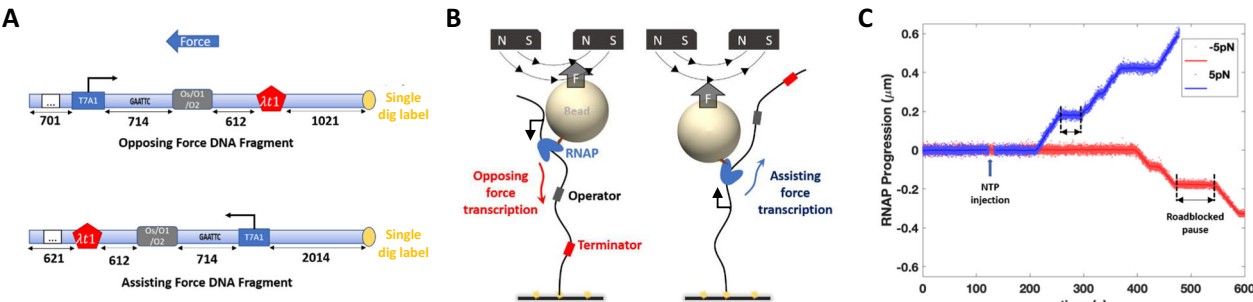

**Fig. 1 | Features of the real-time experiments. A** DNA templates for opposing and assisting force experiments have identical transcribed sequences. The numbers indicate distances in base pairs. **B** A schematic illustration shows force opposing (left) or assisting (right) transcription. **C** Representative records of transcription template length as a function of time under opposing (red) and assisting (blue) force conditions show pauses at LacI roadblock sites (bracketed by black dashed lines).

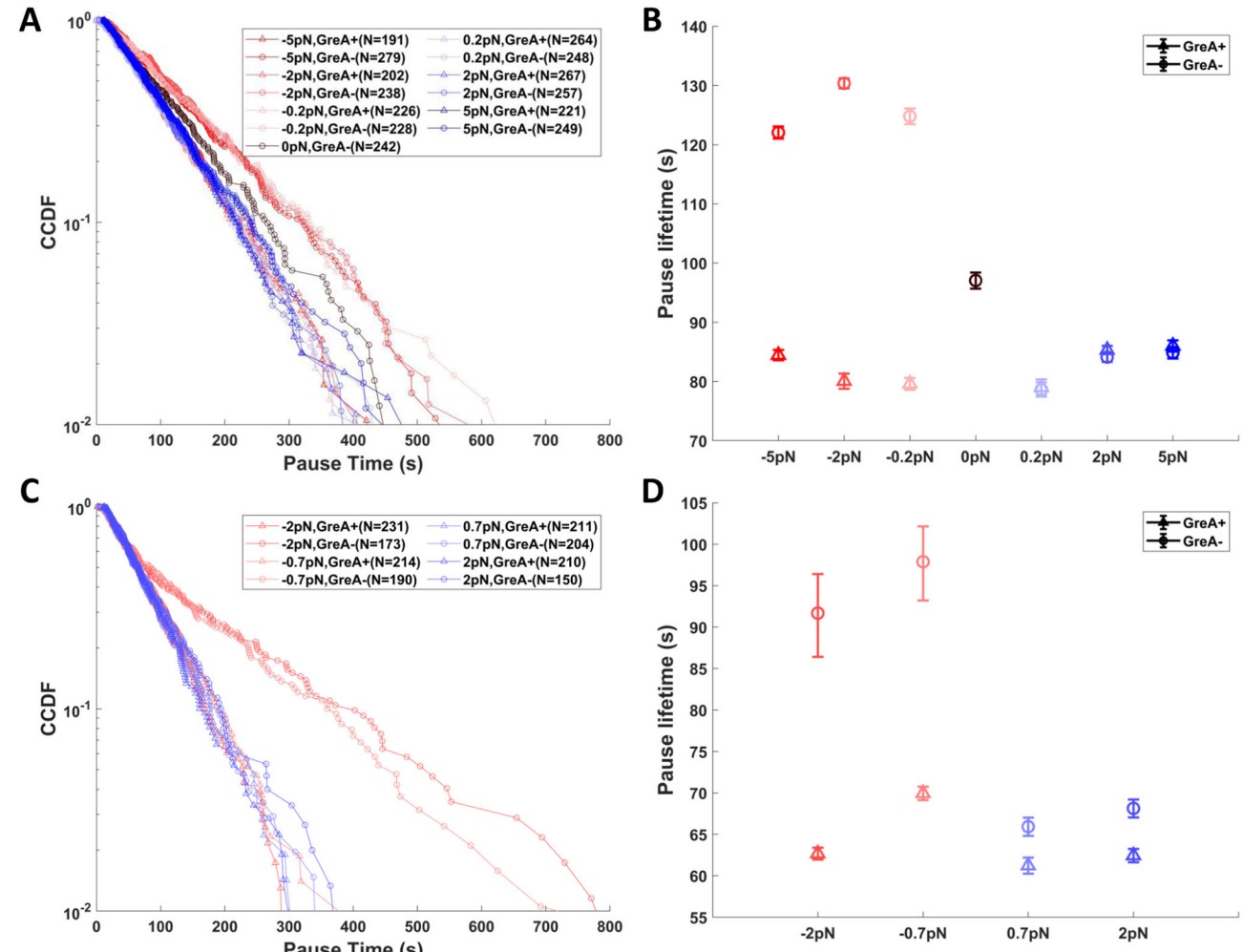

**Fig. 2 | Pauses at LacI-O1 and LacI-O2 roadblocks under conditions of opposing (red) or assisting (blue) force and with (triangles) or without (circles) GreA.** **A** The complementary cumulative distribution function (fraction of pauses longer than a given time, CCDF) and (**B**) characteristic times of pauses at LacI-O1 roadblocks shows that the longest pauses were associated with opposing force without GreA, followed by shorter pauses with no force, and even shorter pauses with assisting force or opposing force with GreA. **C** The CCDF and (**D**) characteristic times of pauses at LacI-O2 roadblocks shows that the longest pauses were associated with opposing force without GreA followed by shorter pauses with assisting force or opposing force with GreA. In (**A**) and (**C**), N represents the number of tethered templates that exhibited a transcription event examined under different force and GreA conditions. Data in (**B**) and (**D**) represent the exponentially fitted characteristic times ± the 90% confidence intervals (Supplementary Figs. 4, 5).

batter through relatively slowly dissociating roadblocks, with GreA accelerating repeated collisions between RNAP and the roadblock. This could be particularly important if interactions with RNAP were to delay dissociation of the roadblock. Such an interaction was not anticipated, but *lac* repressor has been reported to bind to RNAP[22], and this interaction was confirmed by co-partitioning (Supplementary Fig. 1C).

To test this hypothesis, transcription assays were performed against another strong roadblock, the mutant EcoRI Q111 endonuclease which does not interact with RNAP. The protein binds with high affinity but does not cut 5′-GAATTC sites. As expected, EcoRI Q111 successfully blocked nearly 100% of ECs in 50 mM [K+] buffer conditions independently of the force. The addition of GreA increased the level of passage through this roadblock to 20% only under opposing force (Fig. 4A, 50 mM [K+]). Notably, adding GreA to a roadblocked EC produced passage shortly thereafter (Fig. 4B), suggesting that opposing force and GreA promote transit through these long-lived roadblocks.

Under increased salt concentration (150 mM [K+]), the affinity of EcoRI Q111 protein for the recognition site weakens such that a considerable fraction of ECs passed through roadblocks (Fig. 4A,

150 mM [K+])[23]. Therefore, we measured both the distribution of roadblock-induced pause times and the fraction of transit by ECs. In all conditions except with opposing force and GreA (Fig. 4C), ~30–60% of ECs dissociated from EcoRI Q111 roadblocks under high salt. A trace terminating at the roadblock is shown in Supplementary Fig. 2C, D. Such indefinitely blocked ECs produced long tails in pause distributions, even though the mean, assisting-force pause lifetime was only ~80 seconds, which is shorter than pauses at LacI-O1 and comparable to pauses at LacI-O2 (Fig. 4D). Since the processivity of bacterial ECs[24] and the stability of tethers (Supplementary Fig. 1E) are robust with respect to high monovalent salt concentrations, we postulate that the elevated salinity induced dissociation of ECs stalled at roadblocks, preventing observation of more transit events. The addition of GreA reduced the opposing force pauses to a level below the assisting-force baseline level (Fig. 4D). Opposing force also increased the fraction of ECs transiting through EcoRI Q111 roadblocks, which increased even further with the addition of GreA (Fig. 4A, 150 mM [K+]). The observed pause lifetimes are consistent with estimates of the dissociation constant of EcoRI Q111 under high salinity $K_d = 0.12$ nM, assuming a linear relationship between ($K_a$) and ([$M^+$])[25].

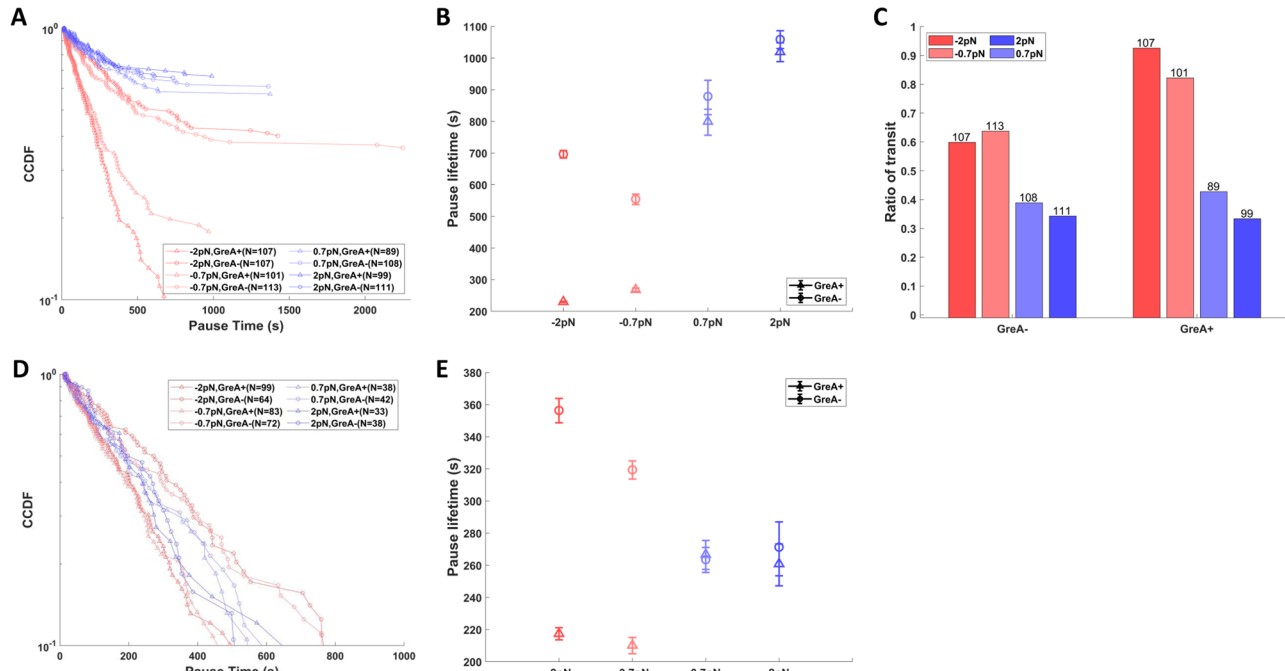

**Fig. 3 | Pause times and fractions of passage at LacI-Os roadblocks under conditions of opposing (red) or assisting (blue) force and with (triangles) or without (circles) GreA. A** The CCDF and (**B**) characteristic times of all recorded pauses at LacI-Os roadblocks show that the longest pauses were associated with assisting force with and without GreA, followed by shorter pauses with opposing force without GreA, and even shorter pauses with opposing force plus GreA. Note that all distributions except that for opposing force plus GreA include a significant fraction of indefinitely paused ECs. **C** Passage through LacI-Os roadblocks was more frequent under opposing than assisting force and was enhanced by the addition of GreA. The number of transcription events in each condition are listed above each bar. **D** The CCDF and (**E**) characteristic times of all except indefinite pauses at LacI-Os roadblocks show that the longest pauses were associated with opposing force without GreA, followed by shorter pauses with assisting force without GreA, and even shorter pauses with opposing force plus GreA. In (**A**) and (**D**), N represents the number of tethered templates that exhibited a transcription event examined under different force and GreA conditions. Data in (**B**) and (**E**) represent the exponentially fitted characteristic times ± the 90% confidence intervals (Supplementary Fig. 6).

A comparison between LacI-O2/O1 induced pauses and LacI-Os/EcoRI Q111 induced pauses shows two clearly different responses of ECs to changes in the direction of force and the addition of GreA. For the LacI-O2/O1 roadblocks, assisting force, either with or without GreA, sets a baseline pause before transit, and GreA must be added to reach that baseline under opposing force. On the contrary, for LacI-Os and EcoRI Q111 roadblocks, opposing force may hasten passage more than the assisting force, especially when GreA is present. To test if the different responses are associated with roadblock strengths, subsets of LacI-Os and EcoRI Q111 data, which exclude the indefinitely stalled ECs, were analyzed. This effectively selects a sub-population of ECs paused at relatively short-lived, lower affinity roadblocks. For such LacI-Os roadblocks, opposing force produced pauses longer than the assisting-force baseline, while addition of GreA shortened the opposing pauses significantly to a level below that baseline (Fig. 3D, E). Similarly, for EcoRI Q111 roadblocks of short duration, the addition of GreA shortened the opposing-force pause time to the level of the assisting-force baseline (Fig. 4E, F), as observed for LacI-O1 and LacI-O2 roadblocks. These results confirm that the lifetime of the roadblock can bias the transit efficiency under opposing forces relative to the assisting force baseline.

The fact that pauses at roadblocks under assisting force are insensitive to GreA indicates that assisting force prevents backtracking. ECs must therefore follow a passive pathway, remaining transcriptionally active, ready to proceed when roadblocks dissociate. For LacI-O1 and LacI-O2, this passive pathway appears to be an efficient transit mechanism; while for LacI-Os and EcoRI Q111 roadblocks, this pathway leads to a significant portion of indefinitely stalled ECs (Figs. 3A, C, 4A, C).

In contrast, GreA significantly enhances transit through roadblocks under opposing force. This finding is consistent with an active, reciprocating pathway that involves backtracking of ECs and GreA-enhanced recovery. Notably, the analyses of both pause lifetimes and fractions of transit suggests that the active pathway speeds transit through LacI-Os and EcoRI Q111 roadblocks (Figs. 3B, C and 4C, D) while slowing transit through LacI-O1 and LacI-O2 roadblocks (Fig. 2B, D). The addition of GreA shortened the opposing force pause times at different roadblocks by different amounts, suggesting that the active pathway may involve multiple cycles of backtracking and recovery before ECs successfully batter through roadblocks.

## A hybrid transit model recapitulates the effects of force and GreA

A hybrid model including the reciprocating/active and passive pathways is consistent with the data. Figure 5A depicts the progression through roadblocks via different states along these pathways. The passive pathway progresses through states ① → ② → ④ → ⑥, and the active pathway through ① → (② → ③ → ②)$_n$ → ⑤ → ⑥ with $n$ cycles of backtracking and recovery. The model includes three kinetic parameters $k_1$, $k_2$ and $k_3$, which represent the backtrack rate, backtrack recovery rate, and roadblock dissociation rate, respectively. Parameter $P1$ represents the probability of dislodging the roadblock at each encounter. Therefore, the transit rate of the passive pathway is simply $k_{passive} = k_3$, and rate of active pathway is $k_{active} = k_1/(1 + k_1/k_2)$.

Notice that ECs likely execute multiple cycles of backtracking and recovery before successfully dislodging a roadblock, which may also spontaneously dissociate during these cycles. Since an EC moves stochastically through the various states, a Monte Carlo simulation is

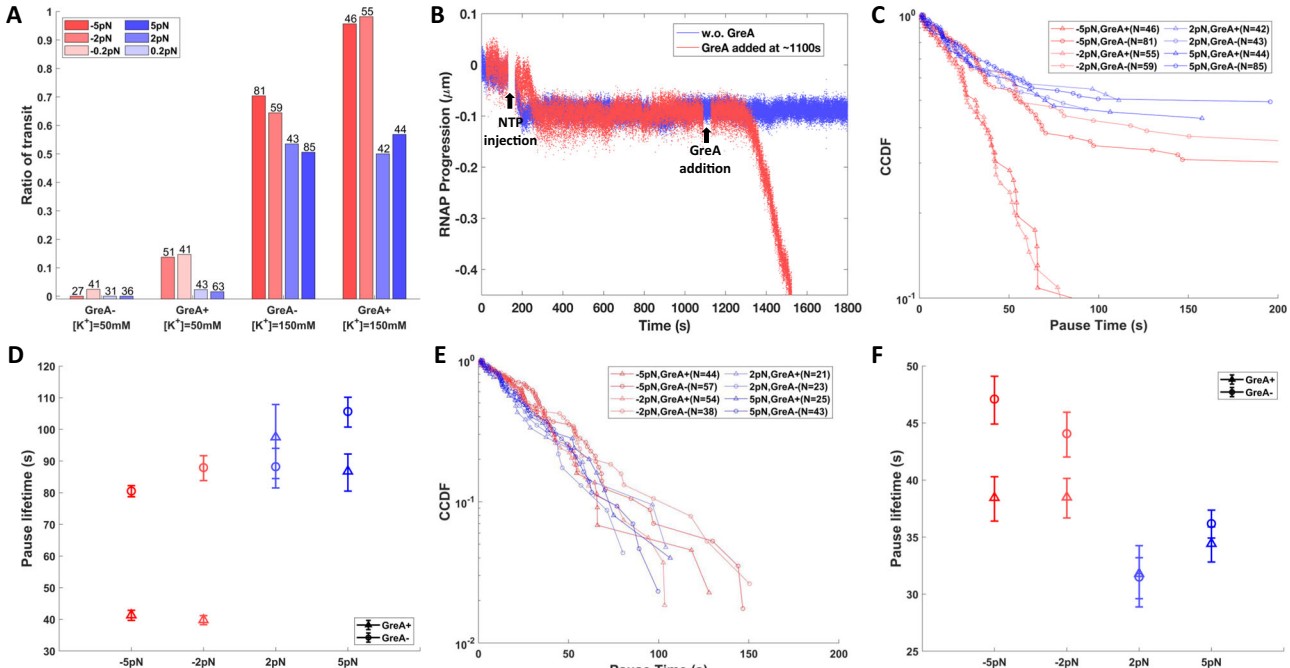

**Fig. 4 | Pause times and the fraction of transit through EcoRI Q111 roadblocks under conditions of opposing (red) or assisting (blue) force and with (triangles) or without (circles) GreA. A** Transit through EcoRI Q111 roadblocks was rare in 50 mM [K+] but increased dramatically in 150 mM [K+] especially upon the addition of GreA. **B** Without GreA most ECs in 50 mM [K+] buffer paused indefinitely at EcoRI roadblocks (blue), but adding GreA (red, ~1100 s) rescued paused ECs that resumed transcription (red, ~1300 s). **C** The CCDFs and (**D**) characteristic times of all recorded pauses at EcoRI Q111 roadblocks show that the longest pauses were associated with assisting force with and without GreA or opposing force without

GreA, and shorter pauses with opposing force plus GreA. Note that all distributions except that for opposing force plus GreA include a significant fraction of indefinitely paused ECs. **E** The CCDFs and (**F**) characteristic times of pauses including only ECs that eventually pass through EcoRI Q111 roadblocks in 150 mM [K+] are shown. In (**C**) and (**E**), N represents the number of tethered templates that exhibited a transcription event examined under different force and GreA conditions. Data in (**D**) and (**F**) represent the exponentially fitted characteristic times ± the 90% confidence intervals (Supplementary Fig. 7).

suitable to reproduce distributions of pause times at roadblocks in different conditions (Algorithm 1). Indeed, such a simulation faithfully reproduced the effects of opposing *versus* assisting force and the addition of GreA (Fig. 5B–E, Supplementary Figs. 8, 9). Details of the simulation are described in Methods and Supplementary Figs. 4–7.

The model predicts changes in pause times and transit percentages produced by GreA and changes in the direction of force. The effects of force and GreA are determined by the relative kinetics of RNAP backtracking-recovery cycling ($k_{active}$) and roadblock dissociation ($k_{passive}$). The simulation reveals three distinct regimes. If cycles of backtracking-recovery are much slower than the dissociation of roadblocks (passive route regime: $k_{active} \ll k_{passive}$), an EC nudged into a backtracking-recovery cycle by opposing force, may not finish a cycle before spontaneous dissociation of the roadblock. For such conditions, the model produced similar, relatively short pauses for the assisting/Gre-, assisting/Gre+ and opposing/Gre+ conditions, but relatively long pauses for the opposing/Gre- condition, resembling the results from the LacI-O2, LacI-O1 and the low-affinity sub-population of high salinity EcoRI Q111 measurements (Figs. 2B, D and 4F, Supplementary Fig. 10A). Notably, in this regime, the simulated pause distribution for the opposing/Gre- condition is much better fitted by a double exponential, representing two stochastic processes with distinct rate constants. A similarly long tail in the distribution of pauses at LacI-O2 roadblocks in opposing/Gre- conditions but not in other experimental conditions, further supports the model (Fig. 2D).

When the two rates are comparable (hybrid route regime: $k_{active} \sim k_{passive}$), opposing force could either lengthen dwell times if RNAP is backtracked at the moment of roadblock dissociation, or shorten dwell times if RNAP successfully dislodges a roadblock before it spontaneously dissociates. The simulated pause distribution

(Supplementary Fig. 10B) resembles the distribution observed with EcoRI Q111 roadblocks at high salt and the sub-population of low-affinity LacI-Os roadblocks, for which opposing/Gre- led to longer pauses than in assisting force conditions and opposing/Gre+ led to shorter pauses (Figs. 3E and 4D). In this regime, the active and passive pathways proceed at similar rates, and all distributions were fitted well by a single exponential.

Finally, if the roadblock dissociation rate is relatively slow compared to backtracking and recovery cycling (active route regime: $k_{active} \gg k_{passive}$), the passive pathway is inefficient, the reciprocating/active pathway is more successful, and opposing force favors backtracking-recovery cycles to produce shorter pauses and higher fractions of EC transit, especially with GreA present. The simulated results (Supplementary Fig. 10C) concur with the experimental results for LacI-Os and low salt EcoRI roadblocks for which opposing force produced shorter pauses and higher fractions of transit than assisting force, and GreA further shortened the pauses and further increased the fraction of transit (Figs. 3B, C and 4A).

Supplementary Table 1 summarizes the values of kinetic parameters from fitting the model to the experimental results. As expected, the passive rates of transit reflect the affinities of roadblocks, $k_{passive}(highsaltEcoRI) \sim k_{passive}(O2) > k_{passive}(O1) > k_{passive}(Os)$. Moreover, the model predicts that transit through EcoRI Q111 (high salt) and LacI-Os (indefinite stalls excluded) roadblocks follows a hybrid route, $k_{active}(highsaltEcoRI/Os) \sim k_{passive}(highsaltEcoRI/Os)$, while transit through LacI-O1 and LacI-O2 roadblocks occurs via the passive route, $k_{active}(O1/O2) \ll k_{passive}(O1/O2)$. These results indicate that cycles of backtracking and recovery occur faster when ECs confront EcoRI roadblocks than LacI roadblocks. We postulate that the different DNA sequences upstream of EcoRI and LacI binding sites may contribute to

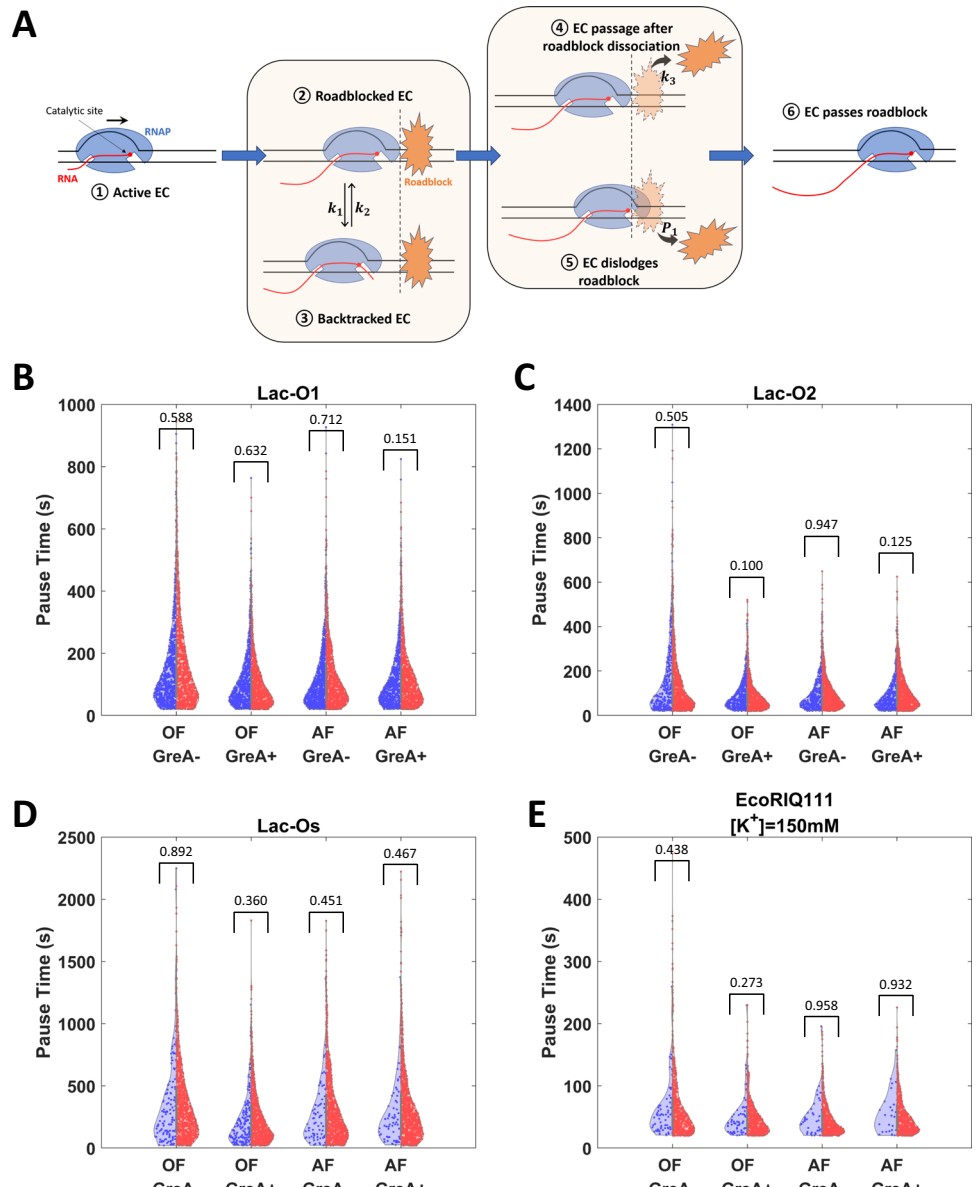

**Fig. 5 | Model and simulation results. A** A proposed model of EC transit through a roadblock includes six states: State ①: Transcription prior to the encounter with roadblock; ②: An EC encounters the roadblock; ③: At a roadblock an EC backtracks with a backtracking rate $k_1$ and recovery rate $k_2$; ④: A roadblock dissociates from DNA spontaneously with a dissociation rate $k_3$; ⑤: An actively transcribing EC, including a recently backtracked EC, has a probability $P_1$ of dislodging the roadblock; ⑥: An EC transits through the roadblock either by actively dislodging the roadblock or after spontaneous dissociation of the roadblock. **B–E** Simulations (red) produced pause time distributions very similar to those observed (blue) in LacI-O1, LacI-O2, LacI-Os and EcoRI Q111 experiments under different conditions. OF and AF represent opposing force and assisting force, respectively. *P* values (>0.05 in all conditions) from two-sided two sample *t*-test are shown in figures.

the difference in the rate of backtracking cycle. Indeed, calculated EC energy profiles are less stable upstream of the EcoRI roadblock, which might suggest faster backtracking and recovery cycles (Supplementary Fig. 1F). Furthermore, the rate of the backtracking and recovery cycle at a LacI roadblock may be slowed by the interaction between RNAP and LacI (Supplementary Fig. 1C), so that battering through only becomes faster than passive dissociation in the case of the LacI-Os roadblock.

The experimental data and the model suggest that the characteristic time of a backtracking-recovery cycle is about 260 s for LacI roadblocks and about 160 s for EcoRI Q111 roadblocks in high salt (Supplementary Table 1), which is longer than the typical lifetime of sequence-induced, backtracked pauses[19,20]. However, similar pauses are frequently observed in a variety of experiments and are classified as stabilized-backtracked pauses[26,27]. Such lengthy, force-independent pauses support the hypothesis that roadblock-induced backtracking might include a force-independent, rate-limiting intermediate state.

As the backtracking recovery rate $k_2$ is likely GreA-concentration dependent, the model predicts that RNAP transit efficiency is affected by GreA levels in the case of reciprocating motion. To further validate the model, we estimated the values of $k_2$ as a function of GreA concentration by assuming a Michaelis-Menten relationship between GreA-faciliated cleavage rate and GreA concentration, and then carried out simulations to generate pause time distributions across varying concentrations of GreA. The simulated pause time dependency on GreA concentration concurred with the experimentally observed characteristic pause lifetimes at various GreA concentrations, lending further support to the proposed model (Supplementary Fig. 11).

## Discussion

Our experimental data and simulations support a model in which RNAP can utilize two mechanisms to bypass obstacles, dynamically shifting between alternative modes of transit through roadblocks of differing strength. In the passive mode, ECs wait and readily proceed upon roadblock dissociation. In the reciprocating mode, ECs execute cycles of backtracking and recovery to batter through roadblocks. Roadblock duration determines which pathway produces more transit. ECs primarily followed a passive route through LacI-O1 and LacI-O2 roadblocks, while the reciprocating backtracking-recovery pathway shortened pauses producing more frequent passage through LacI-Os and EcoRI Q111 roadblocks. Rather than impeding progress, backtracking significantly helps ECs evict DNA-bound proteins that dissociate relatively slowly making the passive route unproductive.

Unexpectedly, roadblocked ECs are exquisitely sensitive to mechanical force and as little as 0.2 pN can significantly change transit. Forces of such magnitude that could impact transcription are prevalent in physiological conditions due to genome anchoring and contacts[28,29]. Therefore, the passage of ECs through roadblocks could be significantly biased by genome architecture and dynamics. However, the transit of ECs through roadblocks was also surprisingly insensitive to the magnitude of forces ranging from 0.2 pN to 5 pN. This contrasts with previous reports in which greater opposing force led to longer backtracked pauses[30,31]. We hypothesize that roadblock-associated backtracking involves a force-sensitive intermediate state followed by a force-insensitive rate-limiting step. Previous studies suggested that external forces bias recovery from backtracking by modulating the forward transcription rate from backtracked positions, whereas bi-directional fluctuations of backtracked ECs were relatively resistant to external forces[20,32]. We speculate that the force-insensitive rate-limiting step is likely the bi-directional diffusion of backtracked ECs, whereas the force-sensitive intermediate is a state in which ECs contact roadblock proteins. The difference between roadblock- and sequence-induced backtracks indicates that there might be functional and mechanistic differences. The former serves to dislodge roadblocks and is easily modulated by mechanical force, while the latter functions as a universal error-correction mechanism and is more resistant to change. The characteristic time of the roadblock-induced backtracking and recovery cycle is predicted to be 160 and 260 s when RNAP confronts LacI and EcoRI Q111 roadblocks respectively. This is significantly longer than the lifetime of sequence-induced backtracks and resembles the dwell time of the "stabilized-backtracked pauses" described elsewhere[27]. Elucidating the mechanisms and functions of backtracked pauses of different lifetimes remains an important subject for future biochemical studies.

Previous studies suggested that backtracked RNAPs contribute to transcription traffic jams. Indeed, *E. coli* employs multiple mechanisms to rescue backtracked RNAPs, such as Gre-factor-dependent RNA cleavage, transcription-translation coupling, and/or trailing RNAPs[6,8,33]. Our study reveals that backtracking may actually have a positive impact, in that an active, reciprocating pathway with backtracking-recovery cycles can promote efficient passage through relatively long-lived roadblocks, which is critical to prevent RNAP from being stalled and targeted by exonucleolytic activity[34]. Our study also explains the different behavior of RNAP and helicase RecBCD when encountering the EcoRI roadblock. The latter can push the EcoRI roadblock thousands of base pairs before finally evicting it, whereas RNAP remains at the roadblock position until passage[35]. In contrast to helicases and translocases that utilize all of the energy of ATP hydrolysis for movement, RNAP, which can only generate chemo-mechanical force during the incorporation of an NTP[36], relies on Brownian-ratchet translocation and produces less chemo-mechanical force on nucleoprotein obstacles. Therefore, repetitive backtracking-recovery cycles are most likely required to perturb and eventually displace high affinity roadblocks.

We note that hindrance by stable protein-DNA contacts is not limited to extrinsic proteins bound to DNA in the path of RNAP. In fact, RNAP faces the same problem every time it initiates transcription: contacts between the promoter DNA elements and the initiation $\sigma$ factor hinder RNAP, triggering repeated synthesis and release of short abortive RNAs or formation of arrested complexes[37]. Gre factors facilitate promoter escape[38], suggesting that cycles of backtracking and RNA cleavage are required to rupture $\sigma$-DNA interactions. Similarly, transcription elongation factors that are recruited to RNAP via DNA, such as RfaH, promote backtracking and depend on Gre factors for escape from the recruitment site[39]. Thus, cycles of backtracking, cleavage, and re-extension are likely to be required for uninterrupted RNA synthesis in all cases when strong DNA-protein contacts, either ahead or behind the moving RNAP, hinder RNA chain extension.

There are other features that are beyond the scope of this work. The experiments and model presented here neglect torsion although reciprocating movement along a few base-pairs in torsionally anchored templates might generate sufficient torsion to stall RNA polymerase[40,41]. Furthermore, backtrack and recovery cycles are faster upstream of EcoRI than upstream of lac binding sites. Sequences with energy profiles spanning a greater range might be used to confirm this effect.

In summary, this study reveals a hybrid mechanism of transit by eleongation complexes through protein roadblocks. ECs can passively wait for dissociation of roadblocks or actively batter through them, and backtracking may lengthen or shorten pauses at roadblocks depending on the longevity of the roadblocks and their interactions with ECs. The effects of tension and the transcript cleavage factor GreA demonstrate that structural (affinity) or dynamic (tension) factors can modulate the efficiency of these pathways and the route followed by ECs to pass through roadblocks. Various elongation factors in vivo might finely tune the efficiency of either pathway making them quite distinct and producing a deterministic response that serves specific biological purposes.

## Methods

### Preparation of proteins

**Biotinylated RNAP holoenzyme.** Biotinylated RNAP holoenzymes were reconstituted from a biotinylated RNAP core enzyme and $\sigma^{70}$ initiation factor, expressed as previously described[42,43].

In brief, *E. coli* BL21 ($\lambda$DE3) harboring pIA1202 ($\alpha$-$\beta$-$\beta'$[AVI][His]-$\omega$) was cultured in LB at 37 °C. The expression was induced at OD600 ~0.5 with 0.5 mM IPTG for 5 h at 30 °C. To achieve better biotinylation, biotin ligase (BirA; Addgene#109424, Watertown, MA) was expressed in *E. coli* BL21 ($\lambda$DE3) in the same way. Cells were pelleted by centrifugation (6000× g, 4 °C, 10 min). Cell pellets were mixed and resuspended in Lysis Buffer (10 mM Tris-OAc pH 7.8, 0.1 M NaCl, 10 mM ATP, 10 mM MgOAc, 100 μM d-biotin, 5 mM $\beta$-mercaptoethanol (ME)) supplied with Complete EDTA-free Protease Inhibitors (Roche Diagnostics, Indianapolis, IN) per manufacturer's instructions.

Cells were sonicated and cell debris was pelleted by centrifugation (20,000 × g, 40 min, 4 °C). The cleared cell extract was incubated with Ni Sepharose 6 Fast Flow resin (Cytiva, Marlborough, MA) for 40 min at 4 °C with agitation. The resin was washed with Ni-A Buffer 25 mM Tris-HCl pH 6.9, 5% glycerol, 150 mM NaCl, 5 mM $\beta$-ME, 0.1 mM phenylmethylsulfonyl fluoride (PMSF) supplemented with 10 mM, 20 mM, and 30 mM imidazole. Protein was eluted in Ni-B Buffer (25 mM Tris-HCl pH 6.9, 5% glycerol, 5 mM $\beta$-ME, 0.1 mM PMSF, 100 mM NaCl, 300 mM imidazole).

The sample was diluted 1.5 times with Hep-A Buffer (25 mM Tris-HCl pH 6.9, 5% glycerol, 5 mM $\beta$-ME) and then loaded onto Heparin HP column (Cytiva). A linear gradient between Hep-A and Hep-B Buffer (25 mM Tris-HCl pH 6.9, 5% glycerol, 5 mM $\beta$-ME, 1 M NaCl) was applied. The biotinylated RNAP core is eluted at ~40 mS/cm.

The elution from Heparin HP column was diluted 2.5 times with Hep-A Buffer and loaded onto Resource Q column (Cytiva). A linear gradient was applied from 5–100% Hep-B Buffer. The biotinylated RNAP core was eluted at ~25 mS/cm.

Fractions from the elution peaks were analyzed by SDS-PAGE. Those containing purified protein were combined and dialyzed against Storage Buffer (20 mM Tris-HCl, pH 7.5, 150 mM NaCl, 45% glycerol, 5 mM $\beta$-ME, 0.2 mM EDTA).

**Lac Repressor Protein.** Lac repressor protein (LacI) was provided by Kathleen Matthews (Rice University)[44].

**EcoRI Q111 Protein.** *E. coli* BL21 ($\lambda$DE3) harboring pVS9 (His6-tagged EcoRI Q111) was cultured in LB at 37 °C. The expression was induced at OD600 ~0.8 with 0.3 mM IPTG for 3 h at 37 °C. Cultures were chilled on ice for 15 min and cells were pelleted by centrifugation (6000 × g, 4 °C, 10 min).

The cell pellet was resuspended in Lysis Buffer (50 mM Tris-HCl, pH 7.5, 1.5 M NaCl, 5% glycerol, 1 mM $\beta$-ME) supplemented with 1 mg/ml lysozyme, 0.1% Tween 20, and Complete EDTA-free Protease Inhibitors (Roche) per manufacturer's instructions. The suspension was incubated on ice for 45 min and sonicated to disrupt cells. Cell debris was pelleted by centrifugation (20,000 × g, 30 min, 4 °C). Cleared extract was incubated with Ni-NTA Agarose slurry (Qiagen, Germantown, MD) for 30 min at 4 °C with agitation. After washing with Wash Buffer (50 mM Tris-HCl, pH 7.5, 0.5 M NaCl, 5% glycerol, 1 mM $\beta$-ME), elution was carried out with 20 mM, 50 mM, 100 mM, and 300 mM imidazole in Heparin Buffer (50 mM Tris-HCl, pH 7.5, 200 mM NaCl, 5% glycerol, 1 mM $\beta$-ME). Fractions containing EcoRI Q111 were pooled and loaded onto Heparin HP column (Cytiva). Protein was eluted by a linear gradient from 200 mM to 800 mM NaCl in Heparin Buffer.

Fractions from the elution peaks were analyzed by SDS-PAGE. Those containing purified protein were combined and dialyzed against Storage Buffer (20 mM Tris-HCl, pH 7.5, 300 mM KCl, 50% glycerol, 0.2 mM DTT, 1 mM EDTA).

**Gre Factor.** Gre factors were purified from plasmids constructed as previously described[45], analyzed by SDS gel electrophoresis, and tested for RNA cleavage activity.

In brief, the Gre expression vector pIA577 was constructed by cloning the *E. coli* gre gene into the NcoI and XhoI sites in the pET28b expression vector (Novagen). For overexpression of the native proteins, pIA577 were transformed into *E. coli* strain BL21($\lambda$DE3). Production of Gre factor was induced according to the Overnight Express protocol (Novagen).

For purification of the Gre factor, cells were harvested and resuspended in lysis T buffer (50 mM Tris-HCl pH 6.9, 1.2 M NaCl, 5% glycerol, 1 mM $\beta$-ME, 0.1 mM PMSF) with Complete EDTA-free protease-inhibitor cocktail (Roche), 0.1% Tween 20 and 1 mg/ml lysozyme. The suspension was incubated on ice for 60 min with occasional swirling and followed by a brief sonication to disrupt the cells. The extract was cleared by centrifugation (27,000× g, 15 min at 277 K). The cleared extract was combined with Ni-NTA agarose (Invitrogen) slurry in lysis T buffer and incubated with agitation for 30 min at 277 K. The slurry was poured into a disposable gravity-flow column and drained. The column was washed with ten volumes of lysis T buffer and ten volumes of the same buffer with 10 mM imidazole. Elution was carried out with five volumes of lysis T buffer with 500 mM imidazole. Fractions containing the protein of interest were combined, concentrated to approximately 3 ml using an Amicon Ultra-15 10,000 MW filter and loaded onto a HiLoad Superdex 75 16/60 column (GE Healthcare) using an AKTA Purifier system (GE Healthcare) at 0.5 ml/min. The column was equilibrated and washed with GF buffer (10 mM Tris-HCl pH 7.8, 1 M NaCl, 2 mM dithiothreitol (DTT)). The purified protein was concentrated to approximately 12 mg/ml and remained stable at 277 K

without degradation for several weeks. The N-terminal His6 tag was not removed prior to crystallization. The yield was 50 mg of purified protein per litre of culture.

These purified proteins have been crystallized and used in functional and single molecule studies by many groups.

The purity of RNA polymerase, EcoRI Q111, and GreA used in this study is shown in Supplementary Fig. 1D.

## Transcription Templates for Magnetic Tweezers Assays
All DNA fragments for MT experiments were PCR amplicons from plasmid templates pZV_NI_400, pDM_N1_400, pWX_12_400 or pDM_E1_400[40], single-digoxigenin labeled forward and unlabeled reverse primer pairs (Supplementary Data 1–5), and Q5 Hot Start High-Fidelity 2X PCR Master Mix (New England Biolabs, Ipswich, MA). The transcribed region had the following spacings: Promoter-709 bp-Lac operator-612 bp-Terminator, with an EcoRI binding sequence 5'-GAATTC between the promoter and the operator site, as illustrated in Fig. 1A. For the opposing force experiments, primers 5'-ATCGTTGGGAACCGGAG and 5'-AGCTTGTCTGTAAGCGGATG were used to generate ~3 kbp DNA fragments with 1021 bp between the chamber surface anchor point and the transcription start site. For the assisting force experiments, primers 5'-dig-GCTTGGTTATGCCGGTACTG and 5'-ACGACCTACACCGAACTGAG were used to generate ~4 kbp DNA fragments with 2014 bp between the anchor point and transcription start site. The longer separation in the assisting force DNA fragment reduces adhesion of DNA-attached magnetic beads to the chamber surface at the beginning of transcription. The fragments produced with single-digoxigenin labeled primers generated torsionally unconstrained tethers for the following transcription assays.

## Microchamber Preparation and Assembly of Transcription Tethers
Microchambers were assembled with laser-cut parafilm gaskets between two glass coverslips[46,47]. The volume of a microchamber was about 10 μL. Polyclonal anti-digoxigenin (Roche Diagnostics, catalogue number 11333089001, lot number 66890900) was introduced to coat the inner surface of the chamber at a concentration of 8 μg/mL in PBS for 90 min at room temperature. The surface was then passivated with Blocking buffer (PBS with 1% caesin, GeneTex, Irvine, CA) for 20 min at room temperature. Transcription tethers were assembled by mixing 30 nM of biotinylated RNAP holoenzyme and 3 nM linear DNA template in Transcription Buffer (20 mM Tris glutamate pH 8, 50 mM potassium glutamate, 10 mM magnesium glutamate, 1 mM DTT, 0.2 mg/ml casein) and incubated 20 min at 37 °C. Afterward, 50 μM ATP, UTP, GTP (NewEngland Biolabs, Ipswich, MA), and 100 μM GpA dinucleotides (TriLink, San Diego, CA) were added to the solution and incubated for additional 10 min at 37°C to allow the ternary complex to initiate transcription and stall at the first G in the template. The solution of ternary complex was diluted to a final concentration of 250 pM RNAP:DNA complex, flushed into the passivated microchambers, and incubated for 10 min. Then, 20 μL of streptavidin-coated superparamagnetic beads (diluted 1:100 in Transcription Buffer; MyOneT1 Dynabeads, Invitrogen/Life Technologies, Carlsbad, CA) were flushed into microchambers to attach beads to biotinylated RNAP stalled on the DNA. After 5 min incubation, excess superparamagnetic beads in solution were flushed out with 50 μL Transcription Buffer.

## Magnetic Tweezers Assays
Magnetic Tweezers were used to observe the dynamics of transcribing ECs by recording the real-time changes in bead height. MTs consist of a pair of permanent magnets positioned above the microchamber that can be translated along the optical axis of a microscope to vary the strength of the magnetic field. The magnitude of tension can be calibrated from the lateral Brownian motion of the bead and the length of the DNA tether[48,49]. Supplementary Fig. 1A shows the magnet height to force calibration in our experiments.

Roadblock proteins were diluted to 20 nM (LacI) and 45 nM (EcoRI Q111) for optimal binding. The concentrations of LacI proteins were chosen based on previous looping experiments and produced minimal off-site binding[47,50]. The concentration of EcoRI Q111 was selected by titrating the binding efficiency at different protein:DNA ratios from AFM images of EcoRI Q111:DNA complexes. The selected EcoRI Q111 concentration optimized specific versus non-specific binding as shown in AFM images (Supplementary Fig. 1B). The diluted roadblock proteins were flushed into microchambers and incubated for 10 min at room temperature prior to recording. Excess roadblock proteins were then flushed out of the chamber with 40 µL of Transcription Buffer to limit binding to a single roadblock protein. We also performed experiments in presence of heparin (Supplementary Fig. 3) to confirm that re-association of dissociated (passive pathway) or disrupted (active pathway) roadblock proteins did not affect pause times.

After 3 min of recording, a complete set of NTPs was added into the chambers to resume transcription of CTP-starved ECs. Recordings lasted at least 30 min for LacI-O1/O2 roadblocks, and more than one hour for the observation of indefinitely stalled ECs.

Since high salt concentrations prevent the assembly and promoter escape of transcription complexes, Transcription Buffer with 50 mM [KGlu] was used to assemble halted transcription complexes. Then, for the high salt experiments, NTPs were flushed into microchambers in high salt Transcription Buffer (500 mM [KGlu]), mixing with normal salt Transcription Buffer in which halted ECs had been prepared, to give an overall concentration ~150 mM [KGlu]. GreA, when used, was flushed into microchambers together with NTPs at 10 µM unless otherwise specified.

Since MT assays apply at least ~0.2 pN to sample even at the furthest magnet-to-microchamber distance (Supplementary Fig. 1A), Tethered Particle Motion (TPM) technique[51] was used to measure pause times in a zero-force condition (Fig. 2A, B, 0 pN, GreA- condition). The preparation of DNA templates and their assembly into tethers for beads in microchambers has been described in detail elsewhere[40].

## Data Analysis

Prior to analyzing the data[17], we inspected the recordings to remove intervals with unusual signal fluctuations which may result from buffer addition or temporary tracking failures. Data in these intervals, mostly during the introduction of NTPs or GreA, were replaced with NaN values and discarded from further analysis. Next, we applied a Bayesian step change detection algorithm to the noisy raw data and extracted step-wise changes in tether lengths. The method minimizes the cost function

$$\mathrm{argmin}_y \left( \sum_i (y_i - \hat{y}_i)^2 + \lambda \sum_i |\hat{y}_{i+1} - \hat{y}_i| \right), \qquad (1)$$

where $\hat{y}$ is the smoothed time series and $y$ is the raw time series. The method, which is described in detail elsewhere[52], reliably produced monotonic traces. The smoothed data consists of flat intervals separated by sharp jumps, and is well suited for detecting pauses (Supplementary Fig. 2A).

For TPM a calibration curve was used to convert excursion values to tether lengths[40,51]. For magnetic tweezing the positions of transcribing ECs along DNA templates was calculated from the end-to-end distance of the tether knowing the force on the tether and the worm-like chain parameters of the DNA template. However, since there are small uncertainties in the force magnitude from tether to tether due to variation in bead size and magnetic susceptibilities (Supplementary Fig. 1A), each individual trace was linearly scaled to align the prominent dwell times at promoter, roadblock and terminator positions. For this purpose, we converted the smoothed, monotonically increasing, or decreasing, traces to dwell time histograms. Since the smoothed traces

change in a strictly step-wise manner, the histograms consist of sharp peaks representing the duration at pause sites. Next, we used expansion $a$ and shift $b$ factors to re-scale the histograms, generating a new set of histograms $S' = a * S + b$, to produce the histogram with maximum pause durations at the promoter, roadblock and terminator positions. Since the start point of transcription was set to 0 and always presents a significant pause, the shift factor was effectively zero. The best-fit values of expansion factor $a$ ranged from 0.75 to 1, depending on the force magnitude in the experiments (Supplementary Fig. 2B). Pauses within ± 20 nm of the roadblock sites in the rescaled histograms were treated as roadblock-induced pauses (Supplementary Fig. 2C–F). The pause time CCDFs collected under different experiment conditions were then fitted by exponential curves. The detailed fitting results were plotted in Supplementary Figs. 4–7.

## Model Analysis

We used the Monte Carlo method to simulate pause times at roadblocks. The algorithm is shown in Algorithm 1.

**Algorithm 1.** Simulate RNAP pause time $t_c$ at a roadblock. States ① – ⑥ are explained in Fig. 5A

Require: RNAP state: ①; Roadblock state: on; $t_{max} = 5000s$; $dt = 1s$; $t_c = 0s$
1:   While RNAP state not ⑥ and $t_c < t_{max}$ do
2:       **if** Roadblock state is *on* then
3:           set Roadblock state to *off* with probability $1 - \exp(-k_3 * dt)$
4:       end If
5:       if RNAP state is ① then
6:           set RNAP state to ⑤
7:       else if RNAP state is ② then
8:           if Roadblock state is *on* then
9:               set RNAP state to ③ with probability $1 - \exp(-k_1 * dt)$
10:          else
11:              set RNAP state to ⑥
12:          end if
13:      **els**e if RNAP state is ③ then
14:          set RNAP state to ⑤ with probability $1 - \exp(-k_2 * dt)$
15:      else if RNAP state is ⑤ then
16:          if Roadblock state is *on* then
17:              set RNAP state to ⑥ with proability *P*1, otherwise set state to ②
18:          else
19:              set RNAP state to ⑥
20:          end if
21:      end if
22:      update current time – $t_c = t_c + dt$
23:  end while

Since Algorithm 1 can simulate a pause time distribution from a set of arbitrary values of parameters, we can search for an optimized set of values that minimize the difference between the simulated and experimental pause time distributions. The details of fitting routines are described in Supplementary Figs. 8, 9. The globally fitted values of parameters are shown in Supplementary Table 1. These sets of values, which reflect the relative kinetics of passive and active/reciprocating routes, reproduced the different responses to forces and GreA conditions of RNAP upon encountering roadblocks corresponding to the three regimes predicted by the model (Supplementary Fig. 10).

## Statistics & reproducibility

No statistical method was used to predetermine sample sizes. At the beginning of an experiment, a randomly selected view with approximately twenty or more particles exhibiting movement, which appear to be tethered by DNA and not stuck on the surface, were selected for

tracking. Randomization was inherent, because DNA tethers with active transcription complexes could not be predicted a priori. The residence times of transcription complexes at roadblock sites in every transcription record were included in the initial data set[17]. Then, a threshold of 20 s for roadblock-induced pausing was set based on previous work in the literature in which transcription elongation complexes were observed to pause as long as 10 s[26]. Pauses shorter than this 20 s threshold were excluded using an automated routine. Due to this automation, the investigators were not blinded to conditions during experiments and analysis.

### Reporting summary

Further information on research design is available in the Nature Portfolio Reporting Summary linked to this article.

## Data availability

The data underlying the findings of this study are available in "figshare" at https://doi.org/10.6084/m9.figshare.24782556[17]. Source data are provided with this paper.

## Code availability

The codes used for data analysis, figure generation and model simulation are available in "figshare" together with the transcription records data https://doi.org/10.6084/m9.figshare.24782556[17]. The codes of MT and TPM softwares used for data acquisition in this study will be available upon request. Requests should be addressed to the corresponding author.

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

## Acknowledgements
LacI was a generous gift from Kathleen Matthews, Rice University. Plasmids for these experiments were created by Derrica McCalla and Zuzsanna Vörös. This work was supported by the National Institutes of Health (NIH) grants R01 GM084070 and R35 GM149296 to LF and R01 GM067153 to IA.

## Author contributions
J.Q. developed the experiments, collected and analyzed data, validated the model, and wrote the manuscript. A.C. helped collect EcoRI Q111 roadblock data. W.X. and Y.Y. performed initial experiments and collected the LacI-O1 data in absence of tension. B.W. prepared proteins. I.A. supervised the preparation of proteins, performed co-immunoseparation analysis of the LacI-RNAP interaction, and participated in writing of the manuscript. D.D. designed the plasmids, participated in the writing of the manuscript, and led the project with L.F. L.F. conceived and co-led the project, contributed to writing and revising the manuscript.

## Competing interests
The authors declare no competing interests.

## Additional information
**Supplementary information** The online version contains Supplementary Material available at https://doi.org/10.1038/s41467-024-47531-x.

