## [Peer Review File · Nature Communications]

Reciprocating RNA Polymerase batters through roadblocksReviewer #1 (Remarks to the Author):

In "Reciprocating RNA polymerase batters through roadblocks", Qian and colleagues investigate how roadblocks on the template DNA obstruct elongating bacterial RNA polymerase (RNAP) progression, and how RNAP eventually resumes elongation through such roadblocks. To this end, the authors use single-molecule magnetic tweezers to monitor RNAP elongation dynamics as a function of the force magnitude and direction, i.e. either assisting or opposing. They use different type of roadblocks, such LacI (and either of the three binding sites, i.e. Os, O1 and O2) and EcoR1, and monitor the dynamics of the collision between each roadblock and the RNAP. They show that upon opposing force and addition of GreA (a transcription factor that rescues RNAP that has entered short backtracks, i.e. <2 nt), the RNAP moves much faster through roadblocks than in the presence of assisting force alone, suggesting an important role of RNAP backtracking to resume elongation. The authors propose a model describing such role for GreA.

Overall, the results are novel and of interest to the community. Therefore, these results would warrant publication in Nature Communications, given that the authors take care of several points listed below. Specifically, the Material and Methods and data statistical information are just not acceptable as such.

Main comments:

1. The experimental and data analysis methodologies are way too short to enable experimental reproduction. For example, the magnetic bead type, the force calibration procedure, the conversion from micron to bp, the trace alignment and scaling procedures, exact DNA sequence and GreA concentration are missing. The authors should extensively complete the materials and methods to make the experiments and their analysis reproducible. They should also cite the relevant literature where warranted: for example, they explain that force calibration can be performed by evaluating the lateral fluctuation of the tethered magnetic bead, but they provide no more insight and cite no literature. The whole section must be significantly revised and completed.
2. The whole study lacks any statistical information (number of traces, number of events per distribution, error calculation...). This is not acceptable and must be revised.
3. Throughout the article: how are defined the error bars?
4. The authors propose a model describing their data. In their model, k_2 should be strongly GreA concentration dependent: what happens if GreA concentration (which is informed nowhere by the way) is decreased 5-fold below k_d ?
5. The representation of the data vs model in Figure 5B-E is not sufficient: the authors must fit their model to the data using a global fitting routine and display the fit with the cumulative distribution to show that this model is accurately describes the data.
6. I am not completely convinced by the authors' model, i.e. the RNAP actively dislodging roadblocks through backtracking-recovery cycle. As mentioned by the authors, RNAP is a molecular motor very sensitive to force, and has been shown to translocate forward through a ratchet-mechanism with an energy bias towards the post-translocated state. In consequence, it is very unlikely to be a molecular motor that can actively remove roadblock. However, I can see another model that would justify their data. GreA mediated backtrack recovery enables RNAP to be immediately available for nucleotide addition and forward translocation upon 3'-RNA cleavage to occupy the roadblock binding site rapidly upon its dissociation. In absence of GreA, the RNAP must escape the backtrack state through 1D diffusion, which is much slower, and therefore inefficient to take advantage of the roadblock dissociation and could enable another roadblock protein present in solution to occupy the binding site. The authors used only 40 μ l of buffer to rinse the 10 μ l chamber, which is very little for proteins with a very high affinity, such as LacI on Os binding site and EcoRI (with k_d in the \sim fM range). Therefore, the authors should reproduce the EcoRI experiments in the presence of heparin in the reaction buffer to make sure that residual proteins cannot bind to the DNA template and show that their results are unaffected.

Minor comments

7. Page 3: "A minor population (<10%) of traces had no roadblock-associated pauses, likely due to incomplete binding of roadblock proteins to the DNA tethers in the microchamber." Where is the data showing such behavior? This should be in SI.

8. Figure 2C and following figures: for the case with GreA, I would rather use triangle markers than no marker. hard to see that they overlap with the "without" case at assisting forces. Same with Figure 2D. Panels order in Figure 2 is confusing.

9. Figure 3A: the figure inset is over the data points. please move that all the data are visible.

10. P.6, third paragraph: "We note that a considerable fraction of ECs...". What is the exact fraction and statistical error?

11. Same paragraph: "We postulate that the elevated salinity induced dissociation of ECs at roadblocks, preventing observation of more passage events.". I doubt the conclusion of the authors, as bacterial RNAP elongation complex has been shown to be robust against high monovalent salt concentration (hundreds of millimolar of NaCl or KCl). However, the single dig-antidig attachment strategy may be more sensitive to salt. The authors should verify whether the complex is stable in bulk using the same buffer.

12. P.12, first paragraph: "and helicase RecBCD when encountering the EcoRI roadblock. The latter can push the EcoRI roadblock thousands of base pairs before finally evicting it, whereas RNAP remains at the roadblock position until passage." I am not sure how RecBCD helicase is related to the RNAP in such context. Remove if not clearly explained.

Reviewer #2 (Remarks to the Author):

Manuscript by Qian et al presents an interesting, and nuanced model of bacterial RNAP encounters with static dissociable roadblocks (here LacI and EcoRI(Q111)). Such encounters are the most common events accompanying transcription in cellulose, where transcribing RNAP traverses operon-sized distances over protein-bound genomic DNA. Detailed understanding of these events is crucial for the proper contextual placement of in vitro transcription experiments within the general framework of gene expression in cellular milieu.

Compared to some previously reported single-molecule experimentation with optical tweezers, experimental results presented here are more fine-grained and exhibit greater physical realism. Together they comprise a rather comprehensive array of confirmatory/benchmarking data and novel/exploratory observations, wherein the former makes easier acceptance of the latter. The observation of particular interest is that of backtracking/reactivation cycling by transcribing RNAP being the source of mechano-chemical force needed for efficient transcription initiation and elongation. Backtracking has been long overlooked altogether and its role in gene regulation and DNA damage surveillance only relatively recently gaining appreciation. The manuscript provides solid support, kinetic and thermodynamic, for the view of backtracking/reactivation cycle as an essential on-pathway mechanism for traversing roadblocks during transcription.

Reported experiments are technically sound, and represent state-of-the-art single-molecule experimentation. Resource authentication is adequate, with the sole exception of the LacI, described as "provided by Kathleen Matthews". Not being personally familiar with Lac repressor lore (and on behalf of similarly disadvantaged reader), this reviewer suggests that an appropriate reference describing Kathleen Matthews' prowess in prepping LacI is appended to 4.1.2 section of Material and Methods.

Manuscript is technically competent, and has above average readability; having said that, a few extra punctuation marks wouldn't go amiss in assisting the reader through some particularly convoluted sentences in sections 1, 2.1 and 2.2.

Summary recommendation by this reviewer is to accept the manuscript with minor revisions, as indicated.

Reviewer #3 (Remarks to the Author):

Our understanding of how RNA polymerases bypass roadblocks on DNA remains incomplete. In this manuscript, based on single-molecule magnetic tweezers experiments, Qian et al. report the dynamics of E. Coli RNA polymerase that transits through different protein barriers on DNA under either assisting or opposing forces. Protein barriers were formed by Lac repressor bound at sites with different affinities or the catalytic dead endonuclease EcoRI at its specific recognition site. The authors also investigated how GreA, the major anti-backtracking factor in E. Coli, affected the pausing of RNA polymerase at these roadblocks. There are many aspects of the findings from this study that are important. Elongation complexes blocked by DNA-bound proteins are highly sensitive to mechanical forces that are physiologically relevant. Pauses at roadblocks under assisting forces are insensitive to GreA. In comparison, GreA significantly enhances passage through roadblocks under opposing forces. The data is also consistent with a hybrid model, including the reciprocating and passive pathways. However, in the current version, there is a lack of description of the robustness of the experiments and broader discussion, including other prevailing models of pausing by RNA polymerases.

Specific comments:

1) Can authors provide SDS-PAGE pictures and quantification of the purity of proteins used in this study (RNA polymerase, LacI, EcoRI Q111, and GreA)?

2) It is mentioned in the Materials and Methods (Page 15) that "the concentration of LacI proteins were chosen for optimal binding". The authors also mentioned using AFM imaging to validate there is minimal off-site binding by EcoRI Q111. Can authors define what is "optimal binding"? Have authors titrated the LacI concentrations in the magnetic tweezers experiments? Can authors provide quantification of EcoRI and LacI off-site binding based on AFM imaging? Does LacI show significant off-site binding under the experimental conditions used in this study?

3) The authors mentioned that LacI proteins were from Kathleen Matthews. Can the authors provide the reference that contains LacI purification procedures?

4) How many biological repeats (protein preps) and technical repeats (magnetic tweezers experiments) were carried out for data shown in Figs, 2, 3, and 4? What are the error bars in these figures? How many measurements are the error bars based on?

5) For Fig. 3C and 4A, can authors provide significance values of the differences between different sets, for example, under opposing forces without or with GreA?

6) Can authors discuss their models in the context of previous ones, such as the "three distinct pause states" proposed by the Dekker group ([https://www.cell.com/cell-reports/pdf/S2211-1247\(22\)00513-7.pdf](https://www.cell.com/cell-reports/pdf/S2211-1247(22)00513-7.pdf))?

7) The authors should add the discussion of "Limitations of the study".

REVIEWER COMMENTS

Reviewer #1 (Remarks to the Author):

In “Reciprocating RNA polymerase batters through roadblocks”, Qian and colleagues investigate how roadblocks on the template DNA obstruct elongating bacterial RNA polymerase (RNAP) progression, and how RNAP eventually resumes elongation through such roadblocks. To this end, the authors use single-molecule magnetic tweezers to monitor RNAP elongation dynamics as a function of the force magnitude and direction, i.e. either assisting or opposing. They use different type of roadblocks, such LacI (and either of the three binding sites, i.e. Os, O1 and O2) and EcoR1, and monitor the dynamics of the collision between each roadblock and the RNAP. They show that upon opposing force and addition of GreA (a transcription factor that rescues RNAP that has entered short backtracks, i.e. <2 nt), the RNAP moves much faster through roadblocks than in the presence of assisting force alone, suggesting an important role of RNAP backtracking to resume elongation. The authors propose a model describing such role for GreA.

Overall, the results are novel and of interest to the community. Therefore, these results would warrant publication in Nature Communications, given that the authors take care of several points listed below. Specifically, the Material and Methods and data statistical information are just not acceptable as such.

Main comments:

1. The experimental and data analysis methodologies are way too short to enable experimental reproduction. For example, the magnetic bead type, the force calibration procedure, the conversion from micron to bp, the trace alignment and scaling procedures, exact DNA sequence and GreA concentration are missing. The authors should extensively complete the materials and methods to make the experiments and their analysis reproducible. They should also cite the relevant literature where warranted: for example, they explain that force calibration can be performed by evaluating the lateral fluctuation of the tethered magnetic bead, but they provide no more insight and cite no literature. The whole section must be significantly revised and completed.

We prepared a new supplementary information document which provides the requested details about the experiments and data analysis procedures.

2. The whole study lacks any statistical information (number of traces, number of events per distribution, error calculation...). This is not acceptable and must

be revised.

The number of traces has been added to the inset legend in the figures. The number of events equals the number of traces in each experimental condition. The calculation of the error bars is described in the Supplemental Information Figure S4–S7.

3. Throughout the article: how are defined the error bars?

See preceding answer.

4. The authors propose a model describing their data. In their model, k_2 should be strongly GreA concentration dependent: what happens if GreA concentration (which is informed nowhere by the way) is decreased 5-fold below k_d ?

We performed experiments under 2 (~ 4 fold below K_d) and 20 μM (~ 3 fold above K_d) to investigate the relationship between pause times and GreA concentration. The experimental results agree with the model simulation as discussed in the last paragraph of section 2.3 and figure S11.

5. The representation of the data vs model in Figure 5B-E is not sufficient: the authors must fit their model to the data using a global fitting routine and display the fit with the cumulative distribution to show that this model is accurately describes the data.

We revised Figure 5B-E to provide more detail and we describe the global fitting routine in supplementary figures S8 and S9.

6. I am not completely convinced by the authors' model, i.e. the RNAP actively dislodging roadblocks through backtracking-recovery cycle. As mentioned by the authors, RNAP is a molecular motor very sensitive to force, and has been shown to translocate forward through a ratchet-mechanism with an energy bias towards the post-translocated state. In consequence, it is very unlikely to be a molecular motor that can actively remove roadblock. However, I can see another model that would justify their data. GreA mediated backtrack recovery enables RNAP to be immediately available for nucleotide addition and forward translocation upon 3'-RNA cleavage to occupy the roadblock binding site rapidly upon its dissociation. In absence of GreA, the RNAP must escape the backtrack state through 1D diffusion, which is much slower, and therefore inefficient to take advantage of the roadblock dissociation and could enable another roadblock protein present in solution to occupy the binding site. The authors used only 40 μl of buffer to rinse the 10 μl chamber, which is very little for proteins with a very high affinity, such as LacI on Os binding site and EcoRI (with k_d in the fM range). Therefore, the authors should reproduce the EcoRI experiments in the presence of heparin in the reaction buffer to make sure that residual proteins cannot bind to the DNA template and show that their results are unaffected.

The model suggested by this Reviewer indeed coincides with our model in conditions of assisting force. However, such a passive model cannot explain the fact that opposing force with GreA can accelerate the passage, which indicates that RNAP reciprocation is critical at least under opposing force.

We have performed measurements in the presence of heparin to prevent the *lac* repressor protein from re-binding to DNA (Figure S3B) and found no difference in the distribution of pause times at the roadblock binding location. This means that re-binding of residual and/or dissociated roadblocks is not affecting our data.

Minor comments:

7. Page 3: “A minor population (<10%) of traces had no roadblock-associated pauses, likely due to incomplete binding of roadblock proteins to the DNA tethers in the microchamber.” Where is the data showing such behavior? This should be in SI.

See figure S2 C & D for randomly-selected traces. These show examples of transcription runs with & without roadblock-induced pauses.

8. Figure 2C and following figures: for the case with GreA, I would rather use triangle markers than no marker. hard to see that they overlap with the “without” case at assisting forces. Same with Figure 2D. Panels order in Figure 2 is confusing.

To avoid confusion, we added a reference to Figure 2, panels A & C when first mentioning the exponential distribution of the pause times and refer to panels B & D in the discussion of the sensitivity to force and GreA. We used triangle markers to indicate GreA+ conditions.

9. Figure 3A: the figure inset is over the data points. please move that all the data are visible.

We fixed the problem.

10. P.6, third paragraph: “We note that a considerable fraction of ECs...”. What is the exact fraction and statistical error?

These fractions are shown in Figure 4A. We have referenced this figure and included the range of values in the text.

11. Same paragraph: “We postulate that the elevated salinity induced dissociation of ECs at roadblocks, preventing observation of more passage events.”.

I doubt the conclusion of the authors, as bacterial RNAP elongation complex has been shown to be robust against high monovalent salt concentration (hundreds of millimolar of NaCl or KCl). However, the single dig-antidig attachment strategy may be more sensitive to salt. The authors should verify whether the complex is stable in bulk using the same buffer.

We performed experiments under high salt buffer in absence of NTPs. We have not observed decreased stability of tether in the high salt conditions (Fig. S1E). Although the elongation complex is active in high salt concentrations, we suspect that high salt might induce premature dissociation of ECs from a template when stalled at roadblocks.

12. P.12, first paragraph: “and helicase RecBCD when encountering the EcoRI roadblock. The latter can push the EcoRI roadblock thousands of base pairs before finally evicting it, whereas RNAP remains at the roadblock position until passage.” I am not sure how RecBCD helicase is related to the RNAP in such context. Remove if not clearly explained.

We hope to emphasize the importance of the reciprocating behavior of RNAPs to dislodge roadblocks. Unlike other motor enzymes that hydrolyze ATP to unwind the helix, a RNAP hydrolyzes NTPs to unwind the helix **AND** synthesize RNA, and a smaller fraction of the energy of ATP hydrolysis is available to power forward motion. Therefore, unlike helicases that can push roadblocks along the template until the dissociation of the roadblock, ratcheting enzymes like RNAP must batter through strong obstacles. We have stated this more explicitly in the text.

Reviewer #2 (Remarks to the Author):

Summary:

Manuscript by Qian et al presents an interesting, and nuanced model of bacterial RNAP encounters with static dissociable roadblocks (here LacI and EcoRI(Q111)). Such encounters are the most common events accompanying transcription in cellulose, where transcribing RNAP traverses operon-sized distances over protein-bound genomic DNA. Detailed understanding of these events is crucial for the proper contextual placement of in vitro transcription experiments within the general framework of gene expression in cellular milieu.

Compared to some previously reported single-molecule experimentation with optical tweezers, experimental results presented here are more fine-grained and exhibit greater physical realism. Together they comprise a rather comprehensive array of confirmatory/benchmarking data and novel/exploratory observations,

wherein the former makes easier acceptance of the latter. The observation of particular interest is that of backtracking/reactivation cycling by transcribing RNAP being the source of mechano-chemical force needed for efficient transcription initiation and elongation. Backtracking has been long overlooked altogether and its role in gene regulation and DNA damage surveillance only relatively recently gaining appreciation. The manuscript provides solid support, kinetic and thermodynamic, for the view of backtracking/reactivation cycle as an essential on-pathway mechanism for traversing roadblocks during transcription.

Minor comments:

Reported experiments are technically sound, and represent state-of-the-art single-molecule experimentation. Resource authentication is adequate, with the sole exception of the LacI, described as “provided by Kathleen Matthews”. Not being personally familiar with Lac repressor lore (and on behalf of similarly disadvantaged reader), this reviewer suggests that an appropriate reference describing Kathleen Matthews’ prowess in prepping LacI is appended to 4.1.2 section of Material and Methods.

A reference to the purification is now included (DOI: [10.1021/bi200896t](https://doi.org/10.1021/bi200896t)).

Manuscript is technically competent, and has above average readability; having said that, a few extra punctuation marks wouldn’t go amiss in assisting the reader through some particularly convoluted sentences in sections 1, 2.1 and 2.2.

We have revised these paragraphs to improve readability.

Summary recommendation by this reviewer is to accept the manuscript with minor revisions, as indicated.

Reviewer #3 (Remarks to the Author):

Our understanding of how RNA polymerases bypass roadblocks on DNA remains incomplete. In this manuscript, based on single-molecule magnetic tweezers experiments, Qian et al. report the dynamics of E. Coli RNA polymerase that transits through different protein barriers on DNA under either assisting or opposing forces. Protein barriers were formed by Lac repressor bound at sites with different affinities or the catalytic dead endonuclease EcoRI at its specific recognition site. The authors also investigated how GreA, the major anti-backtracking factor in E. Coli, affected the pausing of RNA polymerase at these roadblocks. There are many aspects of the findings from this study

that are important. Elongation complexes blocked by DNA-bound proteins are highly sensitive to mechanical forces that are physiologically relevant. Pauses at roadblocks under assisting forces are insensitive to GreA. In comparison, GreA significantly enhances passage through roadblocks under opposing forces. The data is also consistent with a hybrid model, including the reciprocating and passive pathways. However, in the current version, there is a lack of description of the robustness of the experiments and broader discussion, including other prevailing models of pausing by RNA polymerases.

Specific comments:

1) Can authors provide SDS-PAGE pictures and quantification of the purity of proteins used in this study (RNA polymerase, LacI, EcoRI Q111, and GreA)?

We provided the SDS-PAGE picture (Fig. S1D) to show the purity of proteins used in this study.

2) It is mentioned in the Materials and Methods (Page 15) that “the concentration of LacI proteins were chosen for optimal binding”. The authors also mentioned using AFM imaging to validate there is minimal off-site binding by EcoRI Q111. Can authors define what is “optimal binding”? Have authors titrated the LacI concentrations in the magnetic tweezers experiments? Can authors provide quantification of EcoRI and LacI off-site binding based on AFM imaging? Does LacI show significant off-site binding under the experimental conditions used in this study?

AFM images for determining EcoRI Q111 binding efficiency are shown in Figure S1B. Since the records that did not show pauses at the expected roadblock positions were easily distinguished from those showing roadblock-induced pauses and excluded from data analysis, slightly unsaturated binding was acceptable. Therefore, we used these AFM images to ensure that EcoRI Q111 proteins at ~50 nM concentration did not lead to excessive off-site binding. In the set of transcription traces, this concentration produced less than 10 percent of traces with no roadblock-induced pauses.

The concentration of LacI used, 20 nM, was determined in titrations to produce looping in tethered particle motion experiments. This concentration was likely to saturate lac repressor binding sites without excessive non-specific binding. The LacI titration is reported in the literature (DOI: 10.1093/nar/gky021) and is cited in the methods. At this concentration there is likely to be very little off-site (non-specific) binding as shown in quantification of LacI off-site binding based on AFM imaging (DOI: 10.1002/pro.3156)

3) The authors mentioned that LacI proteins were from Kathleen Matthews.

Can the authors provide the reference that contains LacI purification procedures?

A reference to the purification is now included (DOI: 10.1021/bi200896t).

4) How many biological repeats (protein preps) and technical repeats (magnetic tweezers experiments) were carried out for data shown in Figs, 2, 3, and 4? What are the error bars in these figures? How many measurements are the error bars based on?

The number of preparations of proteins used in these experiments was: lac repressor (1), RNAP (2), GreA (1), EcoRIQ111 (1). The error bars are described in Figure S4-7.

5) For Fig. 3C and 4A, can authors provide significance values of the differences between different sets, for example, under opposing forces without or with GreA?

We calculated two-sided t-test's and show the p-values of these t-test's in Figure S4-S7.

6) Can authors discuss their models in the context of previous ones, such as the "three distinct pause states" proposed by the Dekker group ([https://www.cell.com/cell-reports/pdf/S2211-1247\(22\)00513-7.pdf](https://www.cell.com/cell-reports/pdf/S2211-1247(22)00513-7.pdf))?

The Dekker group did not include a roadblock and this manuscript does not consider elemental pauses, so there are substantial differences in the systems. Nonetheless we have commented on these differences and likely similar states in the second paragraph of the discussion. In particular, the backtracking-recovery cycle times determined herein are similar to the pauses of stabilized backtracked complexes observed by the Decker group.

7) The authors should add the discussion of "Limitations of the study".

We now address the study limitations in the penultimate paragraph of the Discussion section.

Reviewer #1 (Remarks to the Author):

Referee report #2 for "Reciprocating RNA polymerase batters through roadblock" by Qian et al.

I thank the authors for providing a detailed response to my queries. Their answers have very much improved the manuscript. A last point remains to be addressed before I can recommend publication. See below.

(1) In Figure S1A, the authors provide a force calibration of their magnetic tweezers assay. Usually, the force versus magnets position is provide with the magnets distance to the magnetic bead, i.e. the closer the magnets, the higher the force. Here, the force increases with the magnets position, which makes little sense: what is actually the distance to the magnetic bead? This figure needs to be revised.

Reviewer #2 (Remarks to the Author):

The authors have diligently addressed all of my points and those raised by other reviewers. I believe the paper is now acceptable.

Reviewer #3 (Remarks to the Author):

The authors have sufficiently addressed all concerns raised. The revised manuscript is suitable for publication in Nature Communications.

REVIEWER COMMENTS

Reviewer #1 (Remarks to the Author):

Referee report #2 for “Reciprocating RNA polymerase batters through road-block” by Qian et al.

I thank the authors for providing a detailed response to my queries. Their answers have very much improved the manuscript. A last point remains to be addressed before I can recommend publication. See below.

(1) In Figure S1A, the authors provide a force calibration of their magnetic tweezers assay. Usually, the force versus magnets position is provide with the magnets distance to the magnetic bead, i.e. the closer the magnets, the higher the force. Here, the force increases with the magnets position, which makes little sense: what is actually the distance to the magnetic bead? This figure needs to be revised.

We regret the oversight, Our magnet travels from position 0 (furthest) to 20 mm (closest). In the previous version of Figure S1A, we plotted the translator position instead of the distance between the sample and the magnet. To eliminate confusion, we have changed the x -axis to indicate the magnet-to-sample distance in the new version of Figure S1A.

Reviewer #2 (Remarks to the Author):

The authors have diligently addressed all of my points and those raised by other reviewers. I believe the paper is now acceptable.

Reviewer #3 (Remarks to the Author):

The authors have sufficiently addressed all concerns raised. The revised manuscript is suitable for publication in Nature Communications.

Reviewer #1 (Remarks to the Author):

The authors have appropriately answered my query, and therefore I recommend this article for publication.